

# Isopsoralen suppresses receptor activator of nuclear factor kappa-$\beta$ ligand-induced osteoclastogenesis by inhibiting the NF-$\kappa$B signaling

Wanda Zhan[1,2,*], Binjia Ruan[3,*], Hui Dong[2], Chaoyong Wang[1], Shuangshi Wu[1], Hang Yu[1,2], Xiaohang Xu[1,2], Hao Sun[2] and Jun Cai[2]

[1] College of Medicine, Yangzhou University, Yangzhou, Jiangsu, China
[2] Clinical Medical College, Yangzhou University, Yangzhou, Jiangsu, China
[3] Medical School, Nanjing University, Nanjing, Jiangsu, China
[*] These authors contributed equally to this work.

Corresponding authors
Hao Sun, s1361191533@163.com
Jun Cai, jun_cai_spine@163.com

## ABSTRACT

Osteoporosis is a serious systemic metabolic bone system disease.This study aimed to identify the target genes of isopsoralen and the signaling pathways involved in the differential expression of the genes involved in osteoclast differentiation. We hypothesized that isopsoralen may inhibit osteoclast differentiation by blocking the nuclear factor kappa-B (NF-$\kappa$B) signaling pathway and verified our hypothesis through basic experiments. The 3-(4, 5-dimethylthiazol-2-yl)-2, 5-diphenyltetrazolium bromide (MTT) assay was used to detect the effect of isopsoralen on the proliferation and viability of primary mouse bone marrow monocytes (BMMCs). The effect of isopsoralen on receptor activator of nuclear factor kappa-B ligand (RANKL)-induced osteoclast differentiation was determined by using tartrate-resistant acid phosphatase (TRAP) staining. Quantitative real-time PCR (qRT-PCR) and Western blot were used to detect the expression of the related genes and proteins. The Kyoto Encyclopedia of Genes and Genomes (KEGG) pathway of isopsoralen target genes were obtained through comprehensive analysis using the STITCH database, Cytoscape 3.8.2, and R-Studio software. Differentially expressed genes (DEGs) were found in osteoclasts induced by RANKL before and after 3 days using R-Studio, following which KEGG analysis was performed. Next, enrichment analysis was performed on the KEGG pathway shared by the target genes of isopsoralen and the differentially expressed genes during osteoclast differentiation to predict the signaling pathway underlying the inhibition of osteoclast differentiation by isopsoralen. Finally, Western blot was used to detect the effect of isopsoralen on the activation of signaling pathways to verify the results of our bioinformatics analysis. Based on the enrichment analysis of isopsoralen target genes and differentially expressed genes during osteoclastogenesis, we believe that isopsoralen can inhibit RANKL-induced osteoclastogenesis by inhibiting the NF-$\kappa$B signaling pathway.

## INTRODUCTION

Osteoporosis is a common bone metabolic disease, which is mainly characterized by the destruction of bone microstructure, the reduction of bone trabecular number, and the Q2 increase of bone brittleness, and a fracture can occur under the action of minor external forces (*Cherubino et al., 2011*). Osteoporosis can be divided into primary osteoporosis and secondary osteoporosis, among which postmenopausal osteoporosis and senile osteoporosis are more common in primary osteoporosis. Secondary osteoporosis refers to osteoporosis that occurs secondary to certain other diseases or after taking certain drugs, such as diabetes, hyperthyroidism, kidney disease and other systemic metabolic diseases or taking a large amount of drugs such as glucocorticoid and immunosuppressive drugs (*Yong & Logan, 2021*).

Bone remodeling is a tightly coupled process that mainly involves osteoclasts and osteoblasts. Osteoclasts are the only bone-resorbing cells in the human body, and they are essential for bone remodeling (*Boyle, Simonet & Lacey, 2003*). They are multinucleated cells generated from the fusion of bone marrow-derived monocyte-macrophages that are induced in the resorption zone in response to macrophage colony-stimulating factor (M-CSF) and RANKL. RANKL induces the secretion of $H^+$, $Cl^-$, cathepsin K (CTSK) and matrix metalloproteinase (MMP) to dissolve the bone matrix (*Cappellen et al., 2002*; *Lorenzo, 2017*). Once RANKL binds to its cognate receptor RANK, it binds to the adaptor protein tumor necrosis factor receptor-associated factor 6 (TRAF6) (*Tanaka et al., 2005*) and then rapidly triggers a signaling cascade including nuclear factor NF-$\kappa$B and mitogen-activated protein kinases (MAPKs) (*Matsumoto et al., 2000*; *Li et al., 2003*; *Wada et al., 2006*; *Lee et al., 2016*), which then activate c-Fos (*Grigoriadis et al., 1994*). Activated NF-$\kappa$B or c-Fos can induce the activation and expansion of nuclear factor of activated T cell cytoplasmic 1 (NFATc1), thereby initiating the expression of osteoclast-related genes.

Isopsoralen is the main active component of psoralen, a coumarin compound with antibacterial, anti-inflammatory, and antitumor properties (*Wei et al., 2016*; *Lu et al., 2014*). Isopsoralen can promote bone and cartilage formation (*Ge et al., 2019*), and it can stimulate the differentiation of bone marrow mesenchymal stem cells into osteoblasts and inhibit their differentiation into adipocytes (*Wang et al., 2017*). In addition, isopsoralen can inhibit the apoptosis of osteoblasts induced by hydrogen peroxide ($H_2O_2$) (*Li et al., 2019*). *In vivo*, isopsoralen also ameliorates osteoporosis caused by sex hormone deficiency in female and male mice (*Yuan et al., 2016*). This suggests that isopsoralen has potential clinical value in the prevention and treatment of postmenopausal osteoporosis. However, the molecular mechanism of isopsoralen is not clear. Therefore, it is necessary to further elucidate the mechanism of isopsoralen in the treatment of osteoporosis from the perspective of molecular biology.

In recent years, with the development of sequencing technology, a variety of genes and their products can be associated with diseases through analysis and visualization of gene expression profiles (*Zhang et al., 2022*). Gene expression varies greatly in different diseases or different subtypes of the same disease. DEGs (between healthy people and those suffering from a disease) can be identified through bioinformatics, and the key genes and

their related molecular mechanisms can also be predicted based on protein interaction and gene enrichment analysis. Bioinformatic analysis has been widely used to identify disease mechanisms and search for target genes.

In this study, we used bioinformatic analysis to identify the KEGG pathway shared by isopsoralen target genes and differentially expressed genes involved in osteoclast differentiation, and to obtain the core genes. We investigated the potential role of isopsoralen in the inhibition of osteoclastogenesis and the specific molecular mechanisms underlying this process.

## MATERIALS AND METHODS

### Reagents

Isopsoralen (purity >98%, Fig. 1A) and the MTT kit were purchased from Solebao Company (Beijing, China). Alpha-modified Eagle's medium ($\alpha$-MEM), fetal bovine serum (FBS), penicillin, and streptomycin were purchased from Gibco (Rockville, MD, USA). TRIzol reagent was purchased from Tiangen (Beijing, China). The SYBR Green Master Mix was purchased from Imgenex (Littleton, CO, USA). Recombinant M-CSF and RANKL were obtained from R&D Systems (Minneapolis, MN, USA). The TRAP staining kit was purchased from Sigma Aldrich (St. Louis, MO, USA). Anti-P65, anti-phospho-P65, anti-I$\kappa$B$\alpha$, and anti-phospho-I$\kappa$B$\alpha$ antibodies were purchased from Cell Signaling Technology (Danvers, MA, USA). Anti-MMP-9, anti-P50/P105, and anti-phospho-P50/P65 antibodies were purchased from Abcam (Cambridge, UK). Anti-GAPDH antibody was obtained from ABclonal Technology (Wuhan, Hubei, China). Anti-NFATc1 antibody was obtained from AiFang Biological (Changsha, Hunan, China). The anti-CTSK antibody was obtained from Proteintech Group (Wuhan, Hubei, China).

### Ethical use of animal

Male C57BL/6 mice, 4 to 5 weeks-old, were purchased from the Comparative Medical Center of Yangzhou University. All the animal experiments were designed and approved by the Yangzhou University Laboratory Animal Ethics Committee (yzu-lcyxy-n059). All animals were raised in the Yangzhou University Center for Comparative Medicine, with different groups raised in different cages. The experimental environment was managed by the Animal Center: a room temperature of 26 °C and 12 h light-dark cycles were maintained, and the breeding environment was cleaned regularly by professionals. All mice were euthanized with anesthetic at the end of the experimental. After freezing treatment, the experimental animal carcasses were sealed and handed over to the Animal Carcass Processing Center of Yangzhou University for centralized harmless treatment. There were no redundant experimental animals in this experiment.

### Cell culture and cell viability assay

Bone marrow monocytes (BMMCs) were extracted from whole bone marrow of the mice as described previously (*Hu et al., 2008*). In brief, femurs of four to five week-old male C57BL/6 mice were taken. After the bone marrow cavity was exposed, the medium was aspirated with a sterile one mL syringe, and the bone marrow cavity was thoroughly

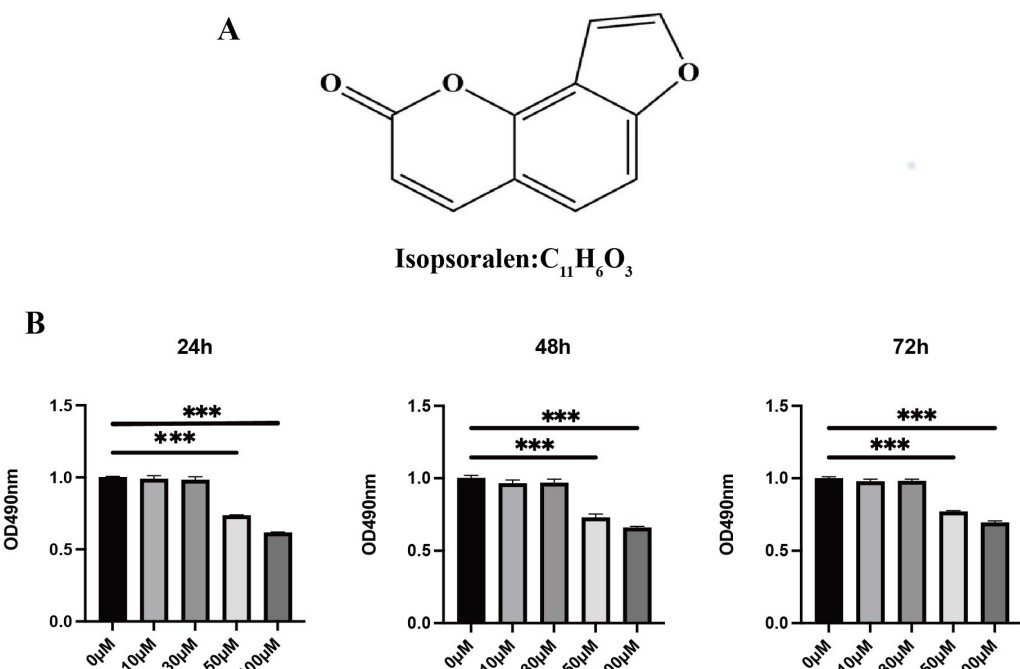

**Figure 1 Effects of isopsoralen on the proliferation of BMMCs.** (A) Chemical structure of isopsoralen. (B) Effects of isopsoralen on the proliferation of BMMCs. The cytotoxic effect of isopsoralen was evaluated using the MTT method. BMMCs were cultured for 24, 48 and 72 h in the presence or absence of varying concentrations of isopsoralen. Optical density was measured at 490 nm. *** $p < 0.001$ in comparison with the negative control group. All assays were repeated three times in triplicate.

flushed until it turned white. After collecting the bone marrow cavity rinse solution, the rinse solution was filtered into a new 50mL centrifuge tube using a 100 μm filter. After centrifugation at 2000 rpm for 5 min, the cells were resuspended using primary osteoclast medium supplemented with 30ng/mL M-CSF (*Yang et al., 2022*; *Liu et al., 2017*), and the cells were evenly spread into a 24-well plate. Unadherent cells were washed with PBS after 48 h, and the remaining cells were BMMCs. BMMCs were supplemented with 10% (v/v) FBS, 1% (w/v) penicillin/streptomycin and 30 ng/mL M-CSF in $\alpha$-MEM complete medium in an incubator at 37 °C and 5% $CO_2$ until ready for further applications. To verify the cytotoxic effect of isopsoralen on BMMCs, cell proliferation and viability were measured using an MTT assay. BMMCs were seeded in 96-well plates at a density of $3 \times 10^3$ cells/well and incubated with different concentrations of isopsoralen (10, 30, 50, and 100 μM) and M-CSF (30 ng/mL) for 24, 48, and 72 h. MTT solution (10 μL/well) was added to each well, and cells were incubated for 4 h. The absorbance was measured at 490 nm using a microplate reader (Thermo Fisher Scientific, Waltham, MA, USA).

### *In vitro* osteoclast formation assay

Whether isopsoralen affects the differentiation of BMMCs was observed using TRAP staining. After the BMMCs were extracted into 24-well plates, and the cells had proliferated to 70% confluence, the differentiation of BMMCs was stimulated with 50 ng/mL RANKL
**Table 1  qRT-PCR primer sequences.**

| Primer name | Primer sequences (5′–3′) |
| --- | --- |
| RT-ACTIN-F | GATCATTGCTCCTCCTGAGC |
| RT-ACTIN-R | GTCATAGTCCGCCTAGAAGCAT |
| RT-MMP9-F | CTGGACAGCCAGACACTAAAG |
| RT-MMP9-R | CTCGCGGCAAGTCTTCAGAG |
| RT- NFATc1-F | CAACGCCCTGACCACCGATAG |
| RT- NFATc1-R | GGGAAGTCAGAAGTGGGTGGA |
| RT-CTSK-F | GAAGAAGACTCACCAGAAGCAG |
| RT-CTSK-R | TCCAGGTTATGGGCAGAGATT |

(*Momiuchi et al., 2021*; *Yin et al., 2019*) and cultured with different concentrations (10, 20, and 30 μM) of isopsoralen to generate mature multinucleated osteoclasts. The cells were then fixed with 4% paraformaldehyde for 1 min, washed with sterile double-distilled water, and TRAP incubation solution was added, followed by incubation at 37 °C for 60 min in the dark. Cells that stained dark red, containing three or more nuclei, were mature osteoclasts.

## Quantitative real-time PCR (qRT-PCR)

Real-time quantitative PCR was used to study the expression of specific genes involved in osteoclast differentiation induced by isopsoralen. BMMCs were lysed with TRIzol and total RNA was extracted. One microgram of total RNA was reverse-transcribed into single-stranded cDNA using oligo-dT primers (Promega Corporation, Madison, WI, USA). qRT-PCR was performed using the SYBR Green Master Mix. The relative mRNA levels were consistent with the expression levels of the housekeeping gene, GAPDH. All data were analyzed using the $2^{-\Delta\Delta CT}$ method. Primer sequences were shown in Table 1.

## Western blot

In order to analyze the effect of protein related to osteoclast differentiation, BMMCs were extracted from 24-well plates. After cell proliferation reached 70%, BMMCs differentiation was stimulated by 50 ng/ml RANKL and cultured with isopsoralen at different concentrations (10, 20, and 30 μM). After reaching this time point, cell lysates were collected. Whole cell lysates were separated by SDS-PAGE and transferred to nitrocellulose membranes. Nonspecific binding was blocked with 5% nonfat milk for 1 h, and the membrane was incubated with the appropriate primary antibody overnight at 4 °C. The membrane was washed three times with TBS-Tween (TBST) and incubated with the appropriate HRP-conjugated secondary antibody for 1.5 h. The color was developed and exposed, and the gray value of the bands was analyzed using Image J software with ECL reagents and a chemoluminescence imaging system.

## Bioinformatics analysis

After predicting the target genes of isopsoralen using the default settings of STITCH, Cytoscape 3.8.2 software was used to construct the interaction network of isopsoralen and its target genes. The enrich KEGG language package in the R-Studio software was used to

enrich and analyze the signaling pathways of isopsoralen target genes. We extracted the data from the GSE176265 database (this database uses RANKL to induce primary mouse mononuclear macrophages, and collects cells for RNA-seq on the 0th, 1st, 2nd, and 3rd days of induction), we converted the gene ID, DEGs were screened according to $P < 0.05$ and |LogFC|>2. KEGG signaling pathway enrichment analysis of DEGs during osteoclast-induced differentiation was performed using R-Studio software. Subsequently, the Venn diagram online tool, Venny 2.1 (http://bioinfogp.cnb.csic.es/tools/venny/index.html) was used to obtain isopsoralen-targeted genes shared with DEGs during the osteoclast-induced differentiation KEGG pathway analysis. The enrichment information for the top five KEGG pathways with the smallest $p$-value was presented using GOplot, an R language package that combines the visualization of expression data with functional analysis. The genes involved in all five KEGG pathways were regarded as core genes. We searched the osteoclast differentiation signaling pathway map on the KEGG official website to identify the signaling pathways involved in the core genes.

## Statistical analysis methods

All basic experiments were performed independently at least three times. SPSS software (version 22.0) was used for statistical analysis, and GraphPad Prism 8 software was used to draw graphs. Quantitative data were described as mean ± standard deviation, independent samples $t$-test was used for comparison between two groups, and one-way analysis of variance was used for comparison between multiple groups. Statistical significance was set to $p < 0.05^*$, $p < 0.01^{**}$, or $p < 0.001^{***}$.

## RESULTS

### Effects of isopsoralen on the proliferation of BMMCs

To probe the effect of isopsoralen on the proliferation and viability of BMMCs, we used an MTT kit to detect cell proliferation and viability (Fig. 1B). The results show that treatment with isopsoralen (<50 μM) for 24, 48, or 72 h did not reduce the viability of BMMCs ($P > 0.05$). When the concentration of isopsoralen was greater than or equal to 50 μM, cell viability decreases gradually with increasing isopsoralen concentration ($P < 0.05$). The results of this experiment determined the maximum concentration of isopsoralen required for the subsequent detection of the effect of isopsoralen on RANKL-induced osteoclast differentiation.

### Isopsoralen inhibits RANKL-induced osteoclast differentiation

To explore whether isopsoralen could inhibit osteoclast differentiation, we plated BMMCs in 24-well plates in the presence of 0, 10, 20, and 30 μM isopsoralen by adding 30 ng/mL M-CSF; BMMCs were treated with 50 ng/mL RANKL. *In vitro* osteoclast differentiation was assessed by TRAP staining after five days, and TRAP-positive cells were quantified. Compared to the RANKL-induced group, isopsoralen reduced the formation of mature osteoclasts in a dose-dependent manner (Figs. 2A and 2B).

To further verify the effect of isopsoralen on osteoclast differentiation, BMMC cells treated as described above. After five days of induction, cell proteins were extracted for

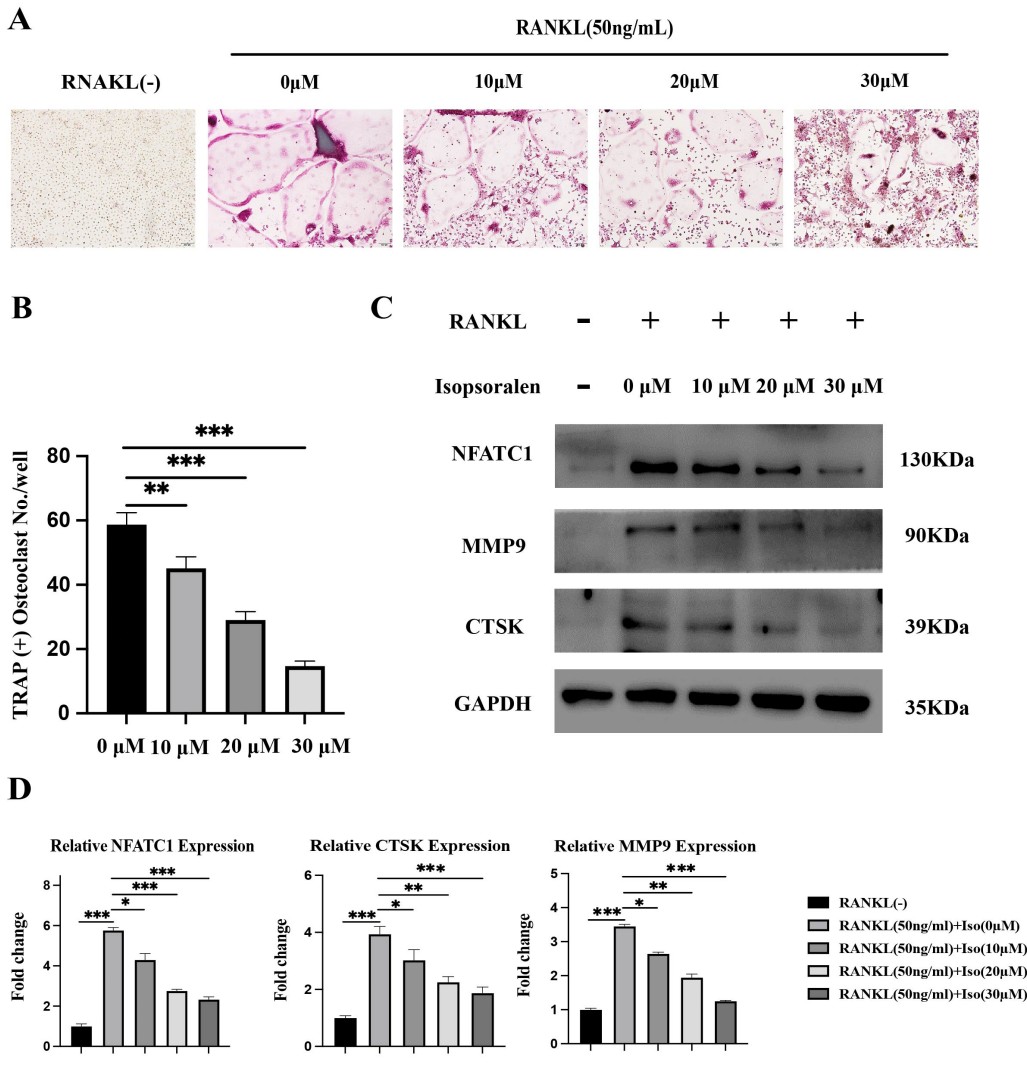

**Figure 2 Isopsoralen inhibits RANKL-induced osteoclast differentiation.** (A, B) Isopsoralen reduces the formation of mature osteoclasts in a dose-dependent manner compared with the RANKL-induced group. (C, D) Effects of different doses of isopsoralen on the protein and mRNA expression of BMMCs with RANKL-induced. * $p < 0.05$, ** $p < 0.01$, *** $p < 0.001$ in comparison with the negative control group. All assays were repeated three times in triplicate. Original magnification, 20×. Scale bar: 100 μm.

Western blot experiments. After five days of induction, isopsoralen had a stronger inhibitory effect on the expression of NFATc1, CTSK, and MMP9 proteins in osteoclasts at increasing concentrations (Fig. 2C). We also extracted cellular RNA to further verify that isopsoralen inhibited the mRNA expression of NFATc1, CTSK, and MMP9 in osteoclasts (Fig. 2D). The results were similar to those of Western blot experiments. Isopsoralen inhibited the expression of osteoclast-related genes induced by RANKL in a concentration-dependent manner.
### Target genes, interaction networks and KEGG analysis of isopsoralen

To explore the mechanism by which isopsoralen inhibits osteoclast differentiation, we used the default search method of the STITCH database to find 30 proteins and compounds that may act on isopsoralen, 28 of which are isopsoralen target genes. Cytoscape 3.8.2 software was used to establish an interaction network (Fig. 3A). The first layer, from the inside to the outside, was the interaction between isopsoralen and protein, and the second layer was the interaction between protein and protein or other compounds. Cytoscape 3.8.2 software analysis obtained the degree, closeness, and betweenness of each target gene in the interaction network. The degree of freedom represents the degree to which a node is related to all other nodes in the network, closeness centrality represents the distance between a node and other nodes in the network (*Yu et al., 2020*), and spacing centrality represents the shortest node between the other two nodes (*Lee, Choi & Cho, 2018*). The next step of visualization and analysis was performed according to the degree of freedom (Fig. 3B) (the darker the color and the larger the circle, the higher the degree of freedom). HDAC1, NF$\kappa$B1, and NF$\kappa$B2 are the three genes with the highest degrees of freedom.

We used the language package in the R-Studio software to perform KEGG signaling pathway enrichment analysis for isopsoralen target genes and draw bubble plots (Fig. 3C). A total of 87 signaling pathways ($P < 0.05$) were enriched. Ranking them from small to large, the top five were the NF-$\kappa$B signaling pathway, Toll-like receptor signaling pathway, C-type lectin receptor signaling pathway, osteoclast differentiation signaling pathway, and Epstein-Barr virus infection signaling pathway.

### DEGs and KEGG analysis of induced differentiation of osteoclasts

We downloaded the GSE176265 dataset (transcriptomic analysis of osteoclastogenesis) from the GEO database and analyzed RNA-Seq data on days 0 and 3 after induction of BMMCs with RANKL through the language package in R-Studio software. After setting the screening criteria at $P < 0.05$, and |LogFC|>2, we obtained 174 differentially expressed genes, of which 111 were downregulated and 63 were upregulated; they were heatmapped using the pheatmap R language package (Fig. 3D).

We used the language package in the R-Studio software to analyze the KEGG signaling pathway enrichment of DEGs during osteoclast-induced differentiation and drew a bubble plot (Fig. 3E). A total of 41 signaling pathways ($P < 0.05$) were enriched. Ranking the *p*-value from small to large, the top five are the neutrophil extracellular trap formation signaling pathway, systemic lupus erythematosus signaling pathway, cell cycle signaling pathway, alcoholism signaling pathway and programmed necrosis signaling pathway.

### KEGG pathway shared by isopsoralen target genes and DEGs during osteoclast-induced differentiation

We obtained 17 isopsoralen target genes and DEGs in the osteoclast-induced differentiation process using the Venn Diagram online tool from the previously obtained isopsoralen target gene KEGG pathway and KEGG pathway of osteoclast-induced differentiation process DEGs. According to the order of *p*-value from small to large (Table 2), the first five were the EB virus infection signaling pathway, TNF signaling pathway, IL-17 signaling pathway,

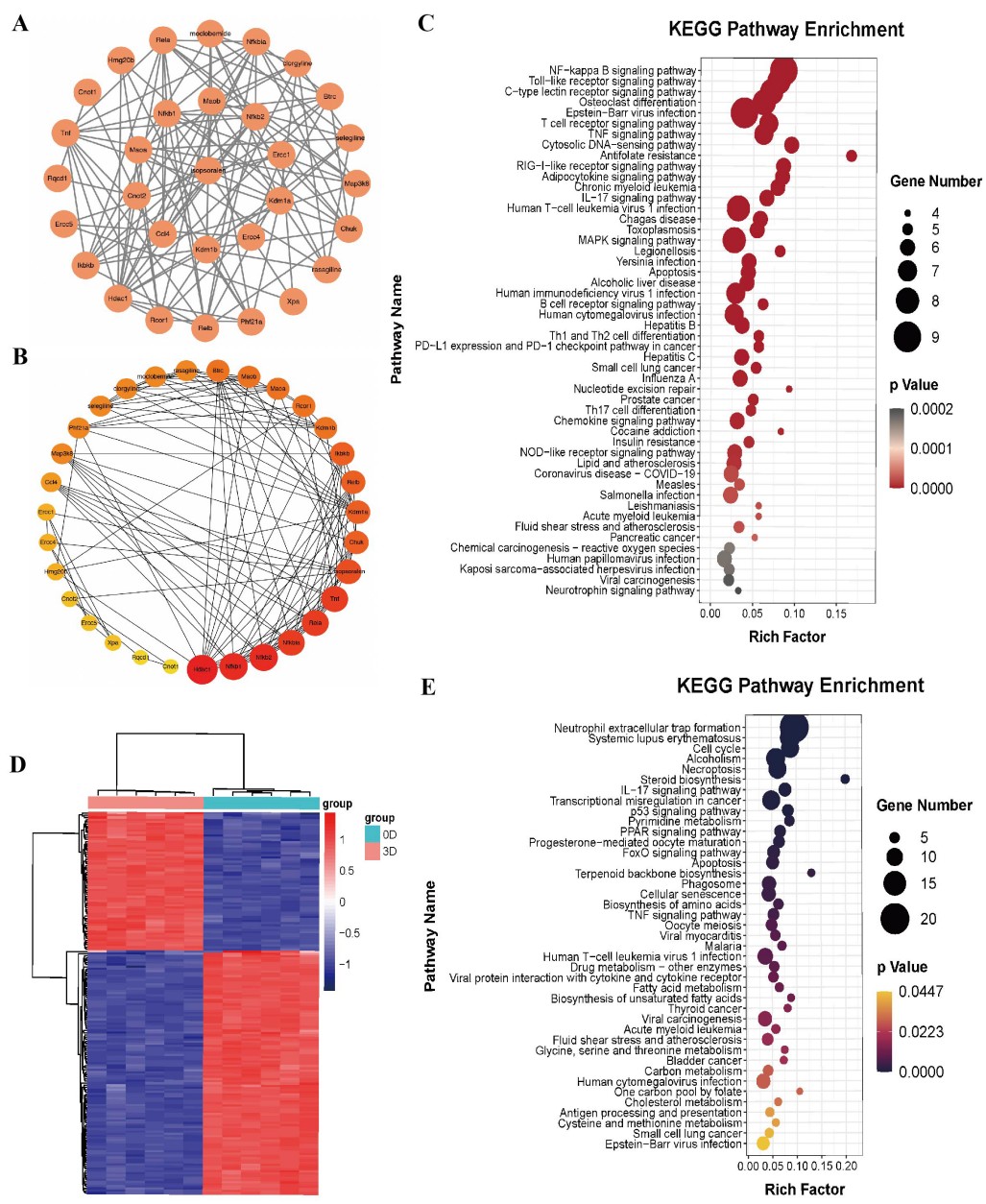

**Figure 3** **Bioinformatics analysis of isopsoralen and DEGs's KEGG analysis of induced differentiation of osteoclasts.** (A) Interaction network of isopsoralen target genes. (B) Visualization of the degrees of freedom of isopsoralen target genes. (C) KEGG analysis of isopsoralen's target gene. (D) Heatmap of DEGs during osteoclast-induced differentiation. (E) KEGG analysis of DEGs during osteoclast-induced differentiation.

**Table 2  Shared KEGG pathways and related genes.**

| KEGG Pathway | Term | Isopsoralen-targeted Genes | $p$ value |
| --- | --- | --- | --- |
| mmu05169 | Epstein-Barr virus infection | IKBKB,TNF,RELA,NFKBIA,RELB,HDAC1,NFKB2,NFKB1,CHUK | $p < 0.001$ |
| mmu04668 | TNF signaling pathway | IKBKB,MAP3K8,TNF,RELA,NFKBIA,NFKB1,CHUK | $p < 0.001$ |
| mmu04657 | IL-17 signaling pathway | IKBKB,TNF,RELA,NFKBIA,NFKB1,CHUK | $p < 0.001$ |
| mmu05166 | Human T-cell leukemia virus 1 infection | IKBKB,TNF,RELA,NFKBIA,RELB,NFKB2,NFKB1,CHUK | $p < 0.001$ |
| mmu04210 | Apoptosis | IKBKB,TNF,RELA,NFKBIA,NFKB1,CHUK | $p < 0.001$ |

human T-cell leukemia virus 1 infection signaling pathway and apoptosis signaling pathway.

### Isopsoralen inhibits RANKL-induced osteoclastogenesis by inhibiting NF-$\kappa$B signaling pathway

We analyzed the top five shared KEGG signaling pathways and related genes through GOplot R package visualization and enrichment analysis to obtain the NFKB1 (P50), TNF, RELA (P65), NFKBIA (I$\kappa$B$\alpha$), IKBKB (IKK–$\beta$), and CHUK (IKK–$\alpha$) genes that are involved in all five KEGG pathways, and therefore, the core genes (Fig. 4A). After bringing the above six core genes into the osteoclast signaling pathway map, we found that the focus was on the NF-$\kappa$B signaling pathway (Fig. 4B); therefore, we predicted that isopsoralen inhibits osteoclasts generation by inhibiting the NF-$\kappa$B signaling pathway.

After RANKL binds to its receptor, RANK, it recruits TRAF6 to its cytoplasmic domain, which subsequently activates signaling pathways such as MAPK and NF-$\kappa$B to induce the expression of osteoclastogenesis genes (*Lee et al., 2005*). We examined the effect of isopsoralen on activation of the NF-$\kappa$B signaling pathway. The analysis of experimental results shows (Fig. 4C) that the levels of p-P65, p-P50, and p-I$\kappa$B $\alpha$ in RANKL-treated BMMCs were significantly increased, while isopsoralen significantly decreased p-P65, p-P65 and p-I$\kappa$B $\alpha$ in a time-dependent manner. These data suggested that isopsoralen inhibits RANKL-induced osteoclastogenesis by inhibiting the NF-$\kappa$B signaling pathway.

## DISCUSSION

Osteoporosis is a global health concern. It is characterized by osteopenia and deterioration of the bone microarchitecture, leading to increased bone fragility and fracture risk (*Chen et al., 2021a*; *Chen et al., 2021b*). Osteoporosis mostly occurs in middle-aged and elderly individuals. As China has entered an aging stage, the number of osteoporosis patients has also increased annually; therefore, it is of great significance to identify relevant diagnostic and treatment actions (*Lin et al., 2015*). Osteoporosis is usually caused by an imbalance in bone metabolism in the body; therefore, osteoporosis treatment is mainly achieved by promoting bone formation and inhibiting bone resorption (*Son et al., 2019*). Many experimental studies have been conducted to treat osteoporosis by promoting the differentiation of bone marrow-derived mesenchymal stem cells into osteoblasts (*Zhi et al., 2021*) and enhancing osteogenic ability (*Chen et al., 2021a*; *Chen et al., 2021b*);

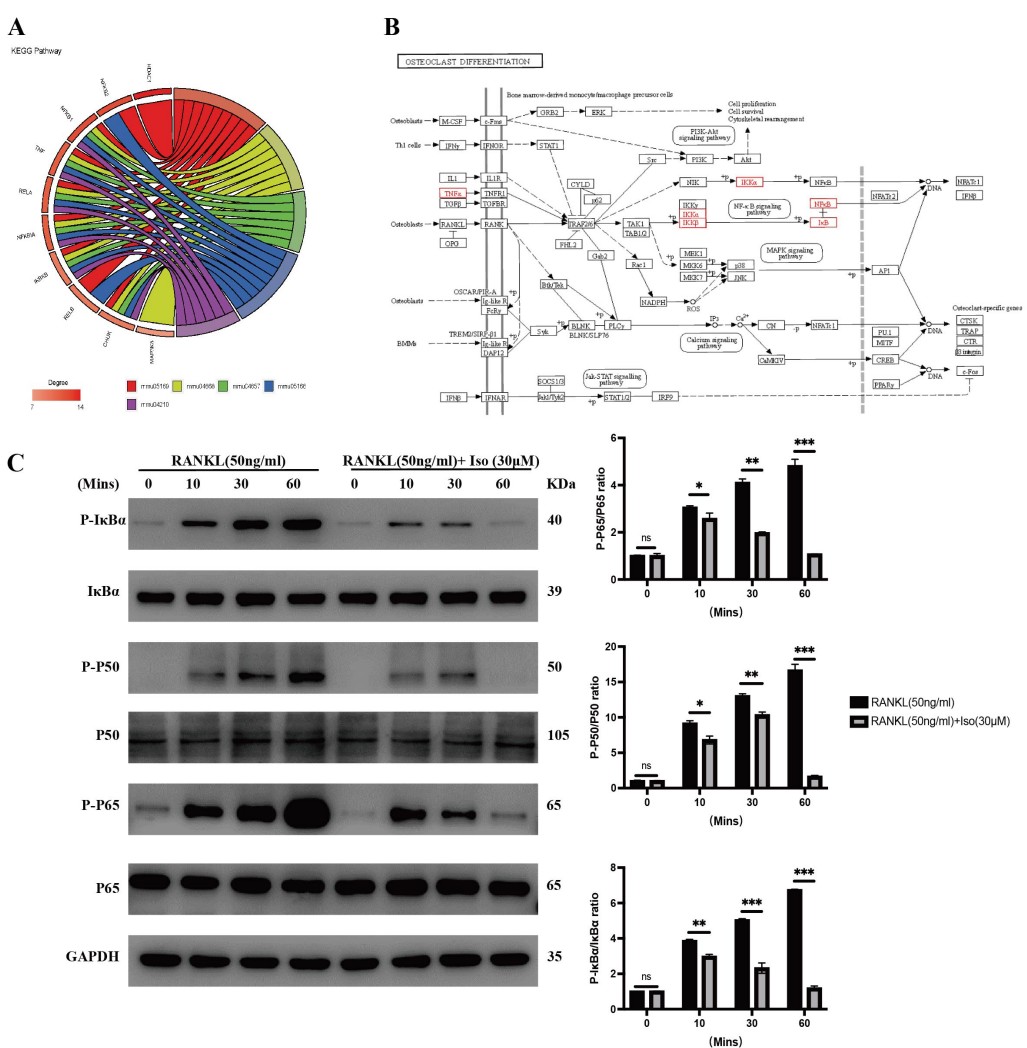

**Figure 4 Isopsoralen inhibits RANKL-induced osteoclastogenesis by inhibiting NF-κB signaling pathway.** (A) Gene enrichment analysis of KEGG signaling pathway. (B) Positional distribution of core genes in osteoclast differentiation signaling pathways. (C) Representative images of Western blot used to analyze the expression of NF-κB pathway related proteins and semiquantification of the data.

however, research on the curative effect of the inhibition of the generation and activation of osteoclasts is still relatively limited. Therefore, it is important to actively search for preventive drugs that can promote bone homeostasis and inhibit the bone loss caused by overactive osteoclasts.

Isopsoralen, an important component of the traditional Chinese medicine, Psoralea, has been extensively studied. Previous research on isopsoralen has focused on its antibacterial, osteogenic, and anti-inflammatory effects (*Li et al., 2018*). However, there are no relevant research reports on the effect of isopsoralen on osteoclastogenesis. Our results demonstrate for the first time that isopsoralen does not inhibit the proliferation and viability of primary mouse mononuclear macrophages when the concentration of isopsoralen is lower than

50 μM, but can significantly inhibit RANKL at increasing concentrations. The induced osteoclasts differentiated, and the transcription and expression of osteoclast-specific genes (NFATc1, MMP9, and CTSK) was significantly inhibited.

Osteoclasts are derived from bone marrow-derived mononuclear macrophages and are the only cells in the body that can absorb bone (*Asagiri & Takayanagi, 2007*). The survival, proliferation, differentiation, and activation of osteoclasts requires the stimulation and induction of M-CSF and RANKL, and the binding of RANKL to its receptor RANK which triggers the fusion and differentiation of osteoclast precursor cells into mature osteoclasts (*Infante et al., 2019*). The gene encoding RANKL is TNFRSF11, and when it is completely absent, mice exhibit severe osteosclerosis with concomitant defects in tooth eruption owing to the complete absence of mature osteoclasts (*Li et al., 2000*). RANKL signaling is also negatively regulated by osteoprotegerin (encoded by TNFRSF11B), a soluble decoy receptor for RANKL that prevents RANKL from binding to RANK (*Yasuda et al., 1998*).

Binding of RANKL to RANK induces trimerization of RANK and recruits the adaptor protein TRAF6 through three TRAF6-binding sites in its C-terminal cytoplasmic tail, thereby initiating downstream signaling cascades such as MAPK, NF-$\kappa$B, and AP-1 (*Wang et al., 2019*). NF-$\kappa$B is a pleiotropic transcription factor; normally, the classical NF-$\kappa$B dimer (P50/RELA) is concentrated in the cytoplasm due to the inhibition of the I$\kappa$B (I$\kappa$B$\alpha$, I$\kappa$B$\beta$, and I$\kappa$B$\varepsilon$) proteins (*An et al., 2019*). TRAF6 can phosphorylate the IKK complex, and the activated IKK complex catalyzes the phosphorylation of I$\kappa$B protein, leading to its ubiquitination. Ubiquitinase degrades I$\kappa$B protein, thereby releasing the NF-$\kappa$B dimer into the nucleus to regulate, in osteoclasts, the expression of specific transcription factors, such as NFATc1 (*Wu et al., 2019*). NF-$\kappa$B signaling is critical for osteoclastogenesis, and P50/P52 double-knockout mice exhibit a pronounced osteosclerotic phenotype due to the failure of osteoclastogenesis (*Franzoso et al., 1997*). Using bioinformatic correlation methods, after enriching the KEGG pathway of isopsoralen target genes and differentially expressed genes involved in osteoclast differentiation, we predicted that isopsoralen might inhibit osteoclasts via the NF-$\kappa$B signaling pathway. Subsequently, we experimentally verified this prediction and demonstrated for the first time that 30 μM isopsoralen can significantly inhibit the phosphorylation of P50, P65, and I$\kappa$B$\alpha$ induced by RANKL, thereby inhibiting the activation of the NF-$\kappa$B signaling pathway and preventing the differentiation and generation of osteoclasts.

There have been extensive studies on the use of isopsoralen in the treatment of osteoporosis, but the mechanism of its therapeutic effect is not fully understood. Through basic experiments and bioinformatics analysis, we demonstrated that isopsoralen can inhibit the differentiation and generation of osteoclasts by inhibiting the activation of the NF-$\kappa$B signaling pathway. Therefore, isopsoralen may be a reference data for treatment of osteoporosis.

## CONCLUSION

Collectively, our study demonstrated that isopsoralen did not affect cell proliferation and viability when the concentration of isopsoralen was less than 50 μM, and that

isopsoralen inhibited the expression of osteoclast-related genes induced by RANKL in a concentration-dependent manner. Our bioinformatics analysis verified that isopsoralen inhibited RANKL-induced osteoclastogenesis by inhibiting the NF-$\kappa$B signaling pathway.

### Funding
This project was funded by the Postgraduate Research & Practice Innovation Program of Jiangsu Province (SJCX22_1817) and the Administration of Traditional Chinese Medicine of Jiangsu Province (ZT202117). The funders had no role in study design, data collection and analysis, decision to publish, or preparation of the manuscript.

### Grant Disclosures
The following grant information was disclosed by the authors:
Postgraduate Research & Practice Innovation Program of Jiangsu Province: SJCX22_1817.
Administration of Traditional Chinese Medicine of Jiangsu Province: ZT202117.

### Competing Interests
The authors declare there are no competing interests.

### Author Contributions
- Wanda Zhan performed the experiments, prepared figures and/or tables, and approved the final draft.
- Binjia Ruan performed the experiments, prepared figures and/or tables, and approved the final draft.
- Hui Dong performed the experiments, prepared figures and/or tables, and approved the final draft.
- Chaoyong Wang performed the experiments, prepared figures and/or tables, and approved the final draft.
- Shuangshi Wu analyzed the data, prepared figures and/or tables, and approved the final draft.
- Hang Yu analyzed the data, authored or reviewed drafts of the article, and approved the final draft.
- Xiaohang Xu analyzed the data, authored or reviewed drafts of the article, and approved the final draft.
- Hao Sun conceived and designed the experiments, authored or reviewed drafts of the article, and approved the final draft.
- Jun Cai conceived and designed the experiments, authored or reviewed drafts of the article, and approved the final draft.

### Animal Ethics
The following information was supplied relating to ethical approvals (*i.e.*, approving body and any reference numbers):

All the animal experiments were designed and approved by the Yangzhou University Laboratory Animal Ethics Committee (yzu-lcyxy-n059).

## Microarray Data Deposition

The following information was supplied regarding the deposition of microarray data:

The data is available in NCBI GEO: GSE176265.

## Data Availability

The raw data are available in the Supplemental File.

## Supplemental Information

Supplemental information for this article can be found online at http://dx.doi.org/10.7717/peerj.14560#supplemental-information.

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
