# Peer review of "Isopsoralen suppresses receptor activator of nuclear factor kappa-β ligand-induced osteoclastogenesis by inhibiting the NF-κB signaling"

_PeerJ, doi:10.7717/peerj.14560_

## Round 0.1 · original submission · Major Revisions

The referees have reviewed your manuscript carefully and recommended some modifications on the manuscript before its further processing. Hence, the decision “Major revision” was taken for your submitted manuscript.

The referee would like to see easily the modifications made to your manuscript in the revised version. Therefore, I invite you to respond to the referees' comments and revise your manuscript carefully.

Do not forget to highlight ALL the changes you make using track changes.

Please provide also an answer/report to the referee(s)’ comments, which summarizes the changes you have made IN the manuscript itself. The answer/report to the referee(s) may also include any other response that you want the editor and the reviewer(s) to note. You should submit the answer/report to the referee(s)’ comments as a separate document.

Thank you for submitting your manuscript to PeerJ and giving us the opportunity to consider your work.

We look forward to receiving your revision.

Reviewer 1 ·

Basic reporting

no comment

Experimental design

no comment

Validity of the findings

no comment

Additional comments

The article is very interesting and well written. It can be accepted after considering the minor revision

Annotated reviews are not available for download in order to protect the identity of reviewers who chose to remain anonymous.

Reviewer 2 ·

Basic reporting

Dear Editor,
The paper entitled " Isopsoralen suppresses RANKL-induced osteoclastogenesis by inhibiting the NF-κB signaling" studies the effect of of isopsoralen on osteoclastogenesis. The overall level of the paper is good, and it is well written. The authors present interesting data and an interesting bioinformatics analysis. Thus, I recommend publication of the manuscript.

Comments to author
In this paper, the authors investigate the importance and potential uses of Isopsoralen in the inhibition of RANKL-induced osteoclastogenesis by inhibiting the NF-»B signaling pathway. In addition, the manuscript is clearly written in professional, unambiguous language. The Introduction section is very short and don’t provide good enough useful information for the readers and also should be more updated and more developed.

Experimental design

A- Justify the use of 6 related proteins in NF-kB pathway on Western blot experiment?
B- Kindly send us the entire row data for the SDS-PAGE, not only the single band for each protein (before and after incubation with the appropriate antibodies)
C- Why did you used M-CSF on the concentration of 30 ng/mL?

Validity of the findings

In the abstract the author mention: we believe, Line 310: we speculate: I think they are not appropriate words.

Additional comments

Please give more details/explanations about the following questions:
A- The authors built his idea on the basis on series of sufficient and reasonable inferences: kindly give more explanations.
B- Justify the use of 6 related proteins in NF-kB pathway on Western blot experiment?
C- The abstract is too long, is more or less same number of lines like introduction section
D- Your introduction needs more detail. I suggest that you improve the description to provide more justification for your study (at lines 59- 66). The introduction section need to be updated (No paper appeared in 2022, 2021 and 2020?)
E- Why NF-kB and not another receptor?
F- Check the list of references.

In the following, there is a list of questions that the authors should consider:
1. Line 65: Space before However
2. Line 67-70: give more details and please added references
3. Line 94: space after Committee
4. Line 107 : correct CO2 to CO2
5. Line 130 ? the symbol is not clear
6. Figure 1 A : correct the formula of the compound (subscript)
7. On figure 1 B: you doubled the amount of isopsoralen used (from 50 to 100µM) but there are slight reduction in the proliferation of BMMCs, Please give an explanation?
8. Line 266-270: kindly added references
9. Line 306 : isostapsene ?
10. Figure 2A: added scale bar and please indicate the magnification of the microscope for each photo

---

## Round 0.2 · accepted · Accept

Dear Dr. Zhan,

Thank you for your submission to PeerJ.

I am writing to inform you that your manuscript - Isopsoralen suppresses RANKL-induced osteoclastogenesis by inhibiting the NF-κB signaling - has been Accepted for publication. Congratulations!

This is an editorial acceptance; publication is dependent on authors meeting all journal policies and guidelines.

Next steps: Your article is being checked and you will receive a list of production tasks shortly. After you complete these tasks, your proofing PDF will be created (please do not proof check your reviewing PDF!).